# Influenza Virus Infections in Cats

**DOI:** 10.3390/v13081435

**Published:** 2021-07-23

**Authors:** Tadeusz Frymus, Sándor Belák, Herman Egberink, Regina Hofmann-Lehmann, Fulvio Marsilio, Diane D. Addie, Corine Boucraut-Baralon, Katrin Hartmann, Albert Lloret, Hans Lutz, Maria Grazia Pennisi, Etienne Thiry, Uwe Truyen, Séverine Tasker, Karin Möstl, Margaret J. Hosie

**Affiliations:** 1Department of Small Animal Diseases with Clinic, Institute of Veterinary Medicine, Warsaw University of Life Sciences—SGGW, 02-787 Warsaw, Poland; 2Department of Biomedical Sciences and Veterinary Public Health (BVF), Swedish University of Agricultural Sciences (SLU), P.O. Box 7036, 750 07 Uppsala, Sweden; sandor.belak@slu.se; 3Department of Biomolecular Health Sciences, Faculty of Veterinary Medicine, University of Utrecht, 3584 CL Utrecht, The Netherlands; H.F.Egberink@uu.nl; 4Clinical Laboratory, Center for Clinical Studies, Department of Clinical Diagnostics and Services, Vetsuisse Faculty, University of Zurich, 8057 Zurich, Switzerland; rhofmann@vetclinics.uzh.ch (R.H.-L.); Hans.Lutz@uzh.ch (H.L.); 5Faculty of Veterinary Medicine, Università degli Studi di Teramo, 64100 Teramo, Italy; fmarsilio@unite.it; 6Maison Zabal, 64470 Etchebar, France; draddie@catvirus.com; 7Scanelis Laboratory, 31770 Colomiers, France; corine.boucraut@scanelis.com; 8Clinic of Small Animal Medicine, Centre for Clinical Veterinary Medicine, LMU Munich, 80539 Munich, Germany; hartmann@lmu.de; 9Fundació Hospital Clínic Veterinari, Universitat Autònoma de Barcelona, Bellaterra, 08193 Barcelona, Spain; albert.lloret@uab.es; 10Dipartimento di Scienze Veterinarie, Università di Messina, 98168 Messina, Italy; mariagrazia.pennisi@unime.it; 11Veterinary Virology and Animal Viral Diseases, FARAH Research Centre, Department of Infectious and Parasitic Diseases, Faculty of Veterinary Medicine, Liège University, B-4000 Liège, Belgium; Etienne.Thiry@ulg.ac.be; 12Institute of Animal Hygiene and Veterinary Public Health, University of Leipzig, 04103 Leipzig, Germany; truyen@vetmed.uni-leipzig.de; 13Bristol Veterinary School, University of Bristol, Bristol BS40 5DU, UK; s.tasker@bristol.ac.uk; 14Linnaeus Group, Shirley, Solihull B90 4BN, UK; 15Institute of Virology, Department for Pathobiology, University of Veterinary Medicine, 1210 Vienna, Austria; karinmoestl@gmail.com; 16MRC—University of Glasgow Centre for Virus Research, Glasgow G61 1QH, UK; margaret.hosie@glasgow.ac.uk

**Keywords:** cats, influenza A virus, low pathogenic, highly pathogenic

## Abstract

In the past, cats were considered resistant to influenza. Today, we know that they are susceptible to some influenza A viruses (IAVs) originating in other species. Usually, the outcome is only subclinical infection or a mild fever. However, outbreaks of feline disease caused by canine H3N2 IAV with fever, tachypnoea, sneezing, coughing, dyspnoea and lethargy are occasionally noted in shelters. In one such outbreak, the morbidity rate was 100% and the mortality rate was 40%. Recently, avian H7N2 IAV infection occurred in cats in some shelters in the USA, inducing mostly mild respiratory disease. Furthermore, cats are susceptible to experimental infection with the human H3N2 IAV that caused the pandemic in 1968. Several studies indicated that cats worldwide could be infected by H1N1 IAV during the subsequent human pandemic in 2009. In one shelter, severe cases with fatalities were noted. Finally, the highly pathogenic avian H5N1 IAV can induce a severe, fatal disease in cats, and can spread via cat-to-cat contact. In this review, the Advisory Board on Cat Diseases (ABCD), a scientifically independent board of experts in feline medicine from 11 European countries, summarises current data regarding the aetiology, epidemiology, pathogenesis, clinical picture, diagnostics, and control of feline IAV infections, as well as the zoonotic risks.

## 1. Introduction

Influenza is a highly contagious, acute infection, usually of the upper respiratory tract, and has been detected worldwide in many vertebrate hosts [1]. Feline respiratory diseases caused by influenza viruses appear to be rather rare and usually self-limiting; however, secondary bacterial infections can lead to complications, and can be associated with fatalities. Very rarely, highly pathogenic influenza viruses can induce a severe, generalised viral disease with a high fatality rate in cats [2].

## 2. Aetiology

The virus is a member of the *Orthomyxoviridae* family. Four types (A, B, C and D) of this agent are known. Influenza virus type A (IAV) is the most important and induces mass disease in humans worldwide, as well as animals, including birds, horses, pigs, minks, ferrets, bats and marine mammals. Dogs and cats may also be affected.

The IAV types are further classified as subtypes based on the antigenicity of the two viral surface proteins, haemagglutinin (H) and neuraminidase (N). There are 16 H and 9 N antigens [1], and their different combinations result in 144 IAV subtypes (e.g., H1N1, H3N8, H5N2, etc.). In addition, further subtypes (H17N10 and H18N11) of influenza-like viruses have been found in bats, but they appear to be distinct from conventional IAVs in multiple aspects [3].

IAVs are genetically highly variable, rapidly changing their antigens, virulence, and ability to replicate in novel host species [4]. Two mechanisms are responsible for this: genetic drift and genetic shift. Genetic drift results from mutations in the genes encoding N or H, producing a new antigenic variant of a given subtype. If the replication of such a variant is less effectively inhibited by the host immunity that eliminated the infection caused by the original strain, the mutated virus can infect the same population again. In contrast, antigenic shift leads to the sudden emergence of a new subtype resulting from the exchange (reassortment) of RNA fragments between two or more IAV subtypes replicating at the same time in a host [4]. Well-recognised “mixing vessel” hosts for human IAVs include pigs and birds, but recent data suggest that dogs and cats might also potentially play such a role [5,6]. These new subtypes, which share pathogenic properties with their parental lineages and have a mixture of the surface antigens of the original strains, can be highly dangerous. As the target host population is often immunologically naïve to the new subtype, epidemics, or even pandemics, in different animal species and humans have arisen in the past [1]. Due to further selection pressure (genetic drift), a new subtype can evolve into multiple antigenic variants, grouped into sublineages or clades [7]. All of these genetic variability mechanisms contribute to the permanent circulation of IAVs in avian and mammalian populations.

IAVs are quickly inactivated by UV light, detergents and disinfectants. However, in water, IAV remains infectious for weeks or months depending on the pH, salinity, and temperature [8,9].

## 3. Circulation of IAVs

Among the possible 144 IAV subtypes, the vast majority have been found in waterfowl (especially ducks, geese, and swans), their natural host [10]. Birds, which are often subclinically infected, can also shed IAVs in their faeces, and their seasonal migrations are crucial for the circulation of IAVs worldwide (Figure 1). The virus can overwinter in ice (e.g., that of northern lakes), which allows for the re-emergence of influenza in the following season.

Generally, IAVs isolated from a given species are able to replicate effectively in this host. Thus, the terms “human strains”, “avian strains”, “equine strains”, etc., are commonly used. However, their high genetic variability facilitates adaptation to other host species, inducing “novel” influenza outbreaks in them. Examples of this include the well-documented epidemics of human influenza that emerged from pigs [1], the canine outbreaks induced by equine H3N8 IAV [11,12] and, in particular, several outbreaks that were generated by avian IAVs in different mammalian species, including dogs and cats [13,14].

On the other hand, variability allows not only the emergence of new viral strains, but also the disappearance of established ones, such as equine H7N7 IAV, which has not been isolated from horses since 1979 [15], or canine H3N8 IAV, which circulated continuously among dogs in the USA for 15 years, but has only been found on rare occasions since 2016 [4].

## 4. Pathogenesis

Most IAVs induce acute upper respiratory tract infections that are either self-limiting or subclinical [1]. Though secondary bacterial infections or other complications may result from IAV infection, these agents are called “low pathogenic strains”. In humans, horses, pigs, dogs and some other species, they replicate only in the upper respiratory tract, inducing the common “seasonal” flu. After an incubation period of one to a few days, inflammation of the bronchial mucosa leads to necrotic lesions, exudate, and lung hyperaemia. Most cases recover after 1 to 3 weeks but, in some, bacterial complications can result in pneumonia; this occurs particularly in poor conditions, as a result of stress, in the very young as well as the old. The fatality rate is <1% for most infections [16]. A natural feline disease of this type has very rarely been seen [17,18,19]. Experimental infections of cats with low pathogenic IAVs hardly ever induce disease, but subclinical replication usually occurs for a few days [20,21,22].

In contrast, appearing mostly in birds, highly pathogenic IAVs are able to replicate not only in the respiratory tract, but also in the gastrointestinal tract, muscles, heart, brain and other organs, resulting in acute, systemic disease that is often associated with a high fatality rate. These IAVs induce sudden, mass mortality in chickens, turkeys and other fowl, but subclinical infections are also common in wild birds. In mammals, this type of influenza is very rare. However, from 1918 to 1920, the worldwide human “Spanish flu” pandemic, with an estimated 20–40 million deaths, was caused by a highly pathogenic H1N1 IAV.

The highly pathogenic avian H5N1 IAV can also induce severe generalised disease in cats after an incubation period of 1–2 days [23]. Virus shedding via the respiratory tract and in faeces starts by day 3 post infection and persists for 4 days or longer [24]. After oral—and possibly also respiratory—infection, replication begins, most probably in the gastrointestinal and/or upper respiratory tracts, and then in the lungs, resulting in foci of alveolar damage [25]. The virus eventually reaches the liver, heart, brain, renal glomeruli, adrenal cortex, and sometimes the spleen, pancreas, and large intestine [24,26,27]. In some domestic cats or wild felids, non-suppurative encephalitis and ganglio-neuritis of the intestinal plexus nervosus were observed [24,28]. Multifocal haemorrhages and necroses in different organs and bronchointerstitial pneumonia are responsible for acute mortality [29].

## 5. Epidemiology of IAV Infections in Cats

For a long time, it was believed that cats were resistant to influenza. Today, it is clear, that cats, dogs, ferrets and other carnivores are involved in the worldwide circulation of IAVs [16].

### 5.1. Low Pathogenic IAVs

Early experiments revealed that cats are susceptible to some IAVs isolated from humans, birds and seals, which usually only induce subclinical infections or a mild fever [20,21,22].

Additionally, canine IAV can occasionally be transmitted to cats. The first outbreak of severe influenza in dogs occurred in 2002 in English foxhounds and was caused by equine H3N8 IAV [30]. Serological studies revealed that this agent, adapted to dogs as canine IAV, was circulating among racing greyhounds in the USA from the early 2000s [31]. After an outbreak in Florida, this virus spread to other breeds and regions of the USA, particularly to shelters [11,32,33]. Another cross-species transmission of the H3N8 IAV to dogs was documented during an epidemic of equine influenza in Australia in 2007 [12]. Natural equine H3N8 IAV infection has not been found in cats thus far, but after experimental inoculation, cats do develop the disease, shed the virus, and transmit the infection to other cats via contact [34].

In South Korea and China, around 2004–2005, a H3N2 IAV emerged in dogs, most probably of avian origin, and became enzootic there [13,35]. Since 2015, this agent has been repeatedly introduced to the USA and Canada by dogs rescued from Asian meat production farms, resulting in several outbreaks [16,36]. Cross-species transmission of this virus is possible as, after experimental inoculations, ferrets, guinea pigs and cats have all been infected [37]. Furthermore, natural feline outbreaks with fever, tachypnoea, sneezing, coughing, dyspnoea and lethargy were noted in two shelters [17,19]. In one of these shelters, the morbidity rate was 100% and the mortality rate was 40%. Although cats can be infected via direct dog-to-cat or cat-to-cat transfer, this virus obviously replicates less efficiently in cats than in dogs, as natural feline outbreaks appear to be very rare. Such outbreaks were largely confined to shelters, and the virus does not appear to undergo prolonged transmission in household cats [36].

From 2016 to 2017, an avian H7N2 IAV infected cats in a New York shelter, and quickly spread to other shelters in New York and Pennsylvania, likely via the movement of cats between the shelters [14,38]. The virus was easily transmitted between cats, but not amongst dogs, chickens, or rabbits housed in the same facilities [39]. In total, approximately 500 cats were found to be infected and most experienced mild respiratory illness [40]. One elderly cat with underlying conditions developed severe pneumonia and was euthanised. Additionally, a veterinarian and a shelter worker, both of whom had multiple direct periods of exposure to the cats without using personal protective equipment, became infected and exhibited mild, transient respiratory disease [41].

Experimental inoculations confirmed that cats were susceptible to the human H3N2 IAV that induced the “Hong-Kong” influenza pandemic in 1968 [20]. Furthermore, several studies indicated that, in 2009, cats (and dogs) worldwide could be infected by the H1N1 IAV during the subsequent human influenza pandemic, probably by direct transmission from their owners [6,42,43,44,45,46,47,48]. In Italy, this virus caused an outbreak of respiratory and gastrointestinal disease in a colony of 90 cats, resulting in 25 deaths [18]. Cat-to-cat transmission was suspected [18,49].

There are also reports of occasional influenza cases in cats caused by other IAVs [16,50,51]. A recent study confirmed that the presence of antibodies to IAVs of both avian and human origin is not uncommon in European shelter cats [6]. Antibodies against H1, H3, H5, H7 and H9 were found in their sera.

### 5.2. Highly Pathogenic H5N1 IAV

In Asia, a highly pathogenic H5N1 virus emerged in 1996, which caused a substantial epidemic of “avian flu” with a high mortality rate in poultry at the beginning of the 21st century. Hundreds of millions of poultry were destroyed [52]. Mammals were sporadically affected, including over 860 humans, with a fatality rate of more than 50% [53]. Severe cases in domestic cats were also noted [2,23], as well as in wild felids [28] that were fed, or had other contact with, infected chickens. In one outbreak, tiger-to-tiger transmission was suspected [54]. As this epidemic reached Europe and Africa, incidental feline cases were also seen there [2], as well as subclinical infections [55]. Usually, these were connected to infected wild birds or poultry. Nevertheless, even in areas in which birds are infected with H5N1 IAV, cats are rarely positive by serology or PCR [56,57]. Experimental infections have confirmed that the highly pathogenic H5N1 IAV may induce a severe, fatal disease in domestic cats, and can spread via cat-to-cat contact [23,24,58]. The virus is excreted not only via the respiratory tract, but also in faeces. It should be stressed that the highly pathogenic H5N1 IAV is still circulating in many parts of the world, including Europe. In the first half of 2021, several outbreaks in wild birds or poultry were noted in Finland, Germany, Denmark, Slovakia, Hungary, France, Latvia, and Estonia [59].

In summary, the data presented in this review clearly show that domestic cats are susceptible to natural IAV infections from other species. They result most likely from close contact with infected humans or animals, especially birds. Serological surveys suggest low to moderate rates of seroconversion to low pathogenic seasonal human or animal strains, and sporadic seroconversions to highly pathogenic avian strains. However, IAVs appear to spread inefficiently among feline populations, probably due to their social organisation, which limits the direct cat-to-cat contact that is required for viral transmission. Thus far, feline influenza epidemics have not been recorded, with only rare outbreaks in dense populations such as shelters. Therefore, cats are not considered a reservoir of influenza. In contrast to humans, horses, pigs, bats, dogs and some other species, the adaptation of IAVs to feline hosts has not yet occurred.

## 6. Clinical Signs

Low pathogenic IAVs usually only induce a subclinical infection or mild, self-limiting upper respiratory tract disease with sneezing as well nasal and/or ocular discharge in cats. Very rarely, in shelters or other crowded colonies, secondary bacterial infections may lead to pneumonia, manifesting as a rise in body temperature, tachypnoea, dyspnoea, coughing, lethargy, and fatalities [17,18,19,40].

Though subclinical infection is also possible, highly pathogenic avian H5N1 IAV in cats usually induces severe clinical signs, including a high fever from day 1 post infection and, by day 2, lethargy, protrusion of the nictitating membranes, conjunctivitis, dyspnoea and a high fatality rate. If diffuse haemorrhagic lesions occur, some cats show serosanguinuous nasal discharge and icterus. Convulsions, ataxia or other neurological signs, as well as gastrointestinal symptoms, can also be seen [23,24,26,54,58].

## 7. Pathological Lesions

In cats that died due to low pathogenic human H1N1 IAV infection, histological examinations revealed bronchointerstitial pneumonia, epithelial bronchiolar hyperplasia and alveolar necrosis [18,42,60].

The necropsy of domestic cats or wild felids infected by the highly pathogenic H5N1 IAV showed multifocal pulmonary lesions and petechial haemorrhages in the lungs, heart, thymus, stomach, intestine, tonsils, mandibular and retropharyngeal lymph nodes and liver, as well as a haemorrhagic pancreatitis [27,28]. Microscopically, these were inflammatory and necrotic lesions.

## 8. Diagnosis

In cats showing signs of acute upper respiratory tract inflammation, influenza should be considered if other etiological agents, such as feline herpesvirus and calicivirus, have been excluded. Risk factors include being in a shelter and close contact with humans or animals suffering from influenza. This applies especially when severe acute respiratory disease is seen in a cat that has outdoor access during an outbreak of highly pathogenic avian influenza infections in poultry and/or aquatic wild birds in the region [2].

IAVs can be isolated in tissue culture or embryonated eggs from nasal or oropharyngeal swabs, or—during a post-mortem examination—from pulmonary tissue (and, in the case of highly pathogenic strains, from rectal swabs or faecal samples, affected organs, intestinal content and pleural fluid).

Viral RNA can be detected in nasal swabs by reverse-transcription PCR during the first 4 days of infection.

In subclinical cases, serology (haemagglutination inhibition tests or neutralisation assays) may be useful for the detection of antibodies. A four-fold serum titre increase within 14 days indicates a recent IAV infection. A comparison of serological assays during a screening study for IAV antibodies in cats has been published recently [6].

For dogs and some other animal species, commercial point-of-care tests are offered for the quick detection of IAV antigens in nasal swabs. These assays have not been validated for cats thus far.

## 9. Control

In the case of an influenza outbreak in a cattery, routine isolation and quarantine procedures should be followed to prevent the spread, as cat-to-cat transmission may occur. The upper respiratory tract disease that occurs as a result is usually mild and self-limiting. In rare, complicated cases, symptomatic medication, combined with the control of secondary bacterial infections, should be implemented alongside other procedures used in cats suffering from other acute viral upper respiratory tract diseases. In humans, oseltamivir is commonly used for the treatment or prevention of IAV infections. This antiviral drug has been given to healthy tigers at risk of highly pathogenic H5N1 IAV infection, but there was no evidence of protection [54].

Though it has been shown that a heterologous avian H5N6 IAV vaccine can protect cats against lethal challenge with the highly pathogenic H5N1 virus [58], no commercial vaccines for cats are available at present. The only prophylaxis is the prevention of any contact with poultry or wild birds infected with H5N1 or other highly pathogenic IAVs. The European Commission has therefore recommended that cats be kept indoors in the areas in which outbreaks of H5N1 IAV infection are recorded in poultry or wild birds [61].

Recently, it was shown that a commercial inactivated H3N2 canine IAV vaccine was well tolerated and induced seroconversion in cats [62]. Even if this vaccine was to be licensed for cats, its usage in Europe is not recommended as this virus has been never detected in Europe, and in regions with canine influenza outbreaks infections in cats are very rare.

## 10. Human Risks

Cats are not considered a reservoir of IAVs. To date, cats infected by both low as well as highly pathogenic IAVs appear to be dead-end host species [63]. Only two cases of cat-to-human transmission of a low pathogenic avian H7N2 IAV have been documented thus far [41,51]. These cases occurred in a shelter after the prolonged and unprotected exposure of a veterinarian and a worker to ill cats and their respiratory secretions, which indicates that the risk of cat-to-human transmission is low [39]. On the contrary, infected humans can be the source of feline infections with seasonal IAVs.

Although cats are sensitive to the highly pathogenic avian H5N1 IAV that caused over 860 human cases worldwide, not one human case was derived from cats. Nevertheless, if highly pathogenic IAVs are found or suspected in a cat, the risk of transmission to humans should be reduced by wearing gloves, a mask and goggles, minimising all unnecessary contact, and carrying out disinfection procedures [2].

## Figures and Tables

**Figure 1 viruses-13-01435-f001:**
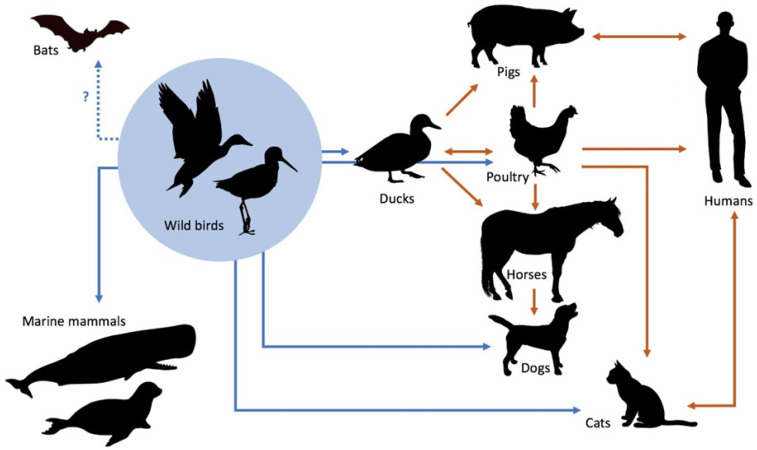
Emergence and transmission of influenza A viruses from aquatic wild bird reservoirs (adapted from [1]).

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
