# Peer review of "Influenza Virus Infections in Cats"

_viruses, 2021, doi:10.3390/v13081435_

Round 1
Reviewer 1 Report
This is an outstanding review of the effects of influenza viruses in mammals and especially in cats. It is particularly relevant today with the many issues and misunderstandings about the origins of coronaviruses and the pandemic of SARS -CoV-2.
Two minor wordsmithing suggestions at the beginning: Line 57. Suggest "---tract, and detected worldwide -- ". Line 66. "----including (or such as) -- " but not both.
Author Response
Thank you for the suggestions.
1. We considered both (line 57 and 66) in the revised version.
2. According to the suggestion of the other reviewer to update the number of hemagglutinin (18) and neuraminidase (11) antigens and give the related reference this fragment (lines 70-74) after revision is as follows: "There are 16 H and 9 N antigens [1], and their different combinations result in 144 IAV subtypes (e.g. H1N1, H3N8, H5N2 etc.). In addition, further subtypes (H17N10 and H18N11) of influenza-like viruses have been found in bats but they appear to be distinct from conventional IAVs in multiple aspects [3]".
[3]. Yang, W,; Schountz, T,; Ma, W. Bat influenza viruses: Current status and perspective. Viruses 2021, 13, 547. doi: 10.3390/v13040547
3. In addition, "inflammation of the bronchial epithelium" has been changed to "inflammation of the bronchial mucosa" (line 127).
Reviewer 2 Report
The Authors wrote a review on Influenza Virus Infection in cats. The manuscript is well written and interesting for readers and worthy of publication. Nevertheless the Authors should update the number of hemagglutinin (18) and neuraminidase (11) and related reference.
Author Response
1. Thank you for the suggestion to update the number of hemagglutinin (18) and neuraminidase (11) and related reference.
This fragment (lines 70-74) after revision is as follows: "There are 16 H and 9 N antigens [1], and their different combinations result in 144 IAV subtypes (e.g. H1N1, H3N8, H5N2 etc.). In addition, further subtypes (H17N10 and H18N11) of influenza-like viruses have been found in bats but they appear to be distinct from conventional IAVs in multiple aspects [3]".
[3]. Yang, W,; Schountz, T,; Ma, W. Bat influenza viruses: Current status and perspective. Viruses 2021, 13, 547. doi: 10.3390/v13040547
2. Two minor wordsmithing suggestions have been corrected according to the other reviewer (lines 57 and 66).
3. In addition, "inflammation of the bronchial epithelium" has been changed to "inflammation of the bronchial mucosa" (line 127).